# Bounding Quality in Diverse Planning

**Michael Katz** and **Shirin Sohrabi** and **Octavian Udrea**

IBM T.J. Watson Research Center
1101 Kitchawan Rd, Yorktown Heights, NY 10598, USA
michael.katz1@ibm.com, {ssohrab,udrea}@us.ibm.com

## Abstract

Diverse planning is an important problem in automated planning with various real world applications. Recently, diverse planning has seen renewed interest, with work that defines a taxonomy of computational problems with respect to both plan quality and solution diversity. However, despite the recent advances in diverse planning the variety of approaches and the number of available tools for these problems are still quite limited, even nonexistent for several computational problems. In this work, we aim to extend the portfolio of approaches and tools for various computational problems in diverse planning. To that end, we introduce a novel approach to finding solutions for three computational problems within diverse planning and present planners for these three problems. For one of these problems, our approach is the first one that is able to provide solutions to the problem. For another, we show that top-k and top quality planners can provide, albeit naive, solutions to the problem and we extend these planners to improve the diversity of the obtained solution. Finally, for the third problem, we show that some existing diverse planners already provide solutions to the problem. Further, we suggest another approach and empirically show that our suggested approach compares favorably with these existing planners.

## Introduction

Diverse planning is an important problem in AI Planning with many practical applications that require generating multiple plans rather than one. Example applications include automated machine learning (Mohr, Wever, and Hüllermeier 2018), risk management (Sohrabi et al. 2018), automated analysis of streaming data (Riabov et al. 2015), and malware detection (Boddy et al. 2005). Diverse planning is also important in the context of re-planning and plan monitoring (Fox et al. 2006), under-specified user preferences (Myers and Lee 1999; Nguyen et al. 2012), as well as plan recognition and its related applications (Sohrabi, Riabov, and Udrea 2016). In all these applications it is important to generate multiple diverse plans, and it is of equal importance to be able to control solution quality.

Most diverse planners developed over the last decade are focused on addressing a particular diversity metric. For example, while the diverse planner DLAMA focuses on finding a set of plans by considering a landmark-based diversity measure (Bryce 2014), other diverse planners such as LPG-d, DIV, DFAA/DFAM, and $A^*AA/A^*AM$ focus on finding a set of plans with a particular minimum action distance (Nguyen et al. 2012; Coman and Muñoz-Avila 2011; Vadlamudi and Kambhampati 2016). Goldman and Kuter (2015) propose a diversity metric based on information retrieval literature. Roberts, Howe, and Ray (2014) suggest another diversity metric, introducing several planners, such as $itA^*$ and MQA, which, in addition to the diversity metrics, consider plan quality. While all these planners implement the chosen diversity metric and switching to another metric is not trivial, the planners DFAA/DFAM and $A^*AA/A^*AM$ work in two phases: finding a set of plans and choosing a proper subset from the found set. That is, selection of a set of plans is independent of the diversity metric. Recently, planners *FI*-diverse were introduced (Katz and Sohrabi 2020). These planners also separate the phase of finding candidate plans from choosing a diverse subset of these plans. Further, the authors provide a tool for selecting a subset of plans for a variety of metrics and computational problems (Katz and Sohrabi 2019).

Another important recent contribution introduced a taxonomy of computational problems and classified existing planners according to the problems they tackle (Katz and Sohrabi 2020). Most existing planners, according to that taxonomy, solve *Satisficing Diverse Planning* (sat-k), where any sufficiently large set of plans is a solution, and the aim is to improve solution diversity. The planners LPG-d (Nguyen et al. 2012) and bFI (Katz and Sohrabi 2020) tackle *Bounded Diversity Diverse Planning* (bD-k), where a set of plans is a solution only if its diversity is above a certain specified bound. Top-k planners (Katz et al. 2018b; Speck, Mattmüller, and Nebel 2020) and top-quality planners (Katz, Sohrabi, and Udrea 2020), while usually are not considered as diverse planners, according to the aforementioned taxonomy return, albeit naive, solutions to the *Bounded Quality Diverse Planning* (bQ-k) problem, where plan set quality is required to be above a specified bound. The planners DFAA/DFAM and $A^*AA/A^*AM$ (Vadlamudi and Kambhampati 2016) tackle *Bounded Quality and Diversity Diverse Planning* (bQbD-k), where both the quality and the diversity of plan sets is bounded from above.

Despite these recent advances in diverse planning, the pool of existing tools is still quite limited. The planners DFAA/DFAM and A*AA/A*AM by Vadlamudi and Kambhampati (2016) are the only existing planners for bQbD-k. Top-quality planners (Katz, Sohrabi, and Udrea 2020), although technically solving bQ-k, do not aim at improving the diversity of the solution. No planners exist for other computational problems, such as *Optimal Diversity Bounded Quality Diverse Planning* (bQoptD-k), where a solution corresponds to a set of plans of best diversity among the sets of bounded quality.

In this work, expand the pool of available planners for diverse planning. We introduce novel planners for the three aforementioned computational problems, bQ-k, bQbD-k, and bQoptD-k, exploiting the recently introduced top-quality planners. To that end, we introduce a novel quality metric that reflects bounded plan costs, making a connection between the costs of plans in a set and quality of a set of plans. Focusing on the most popular diversity metric (Nguyen et al. 2012; Coman and Muñoz-Avila 2011; Vadlamudi and Kambhampati 2016), for all three planners we generate a subset of all plans of bounded quality as a first step. In the second step, we select a subset of plans found in the first step that constitutes a solution to the respective computational problem. For bQ-k, as any sufficiently large subset of plans is, albeit naive, a solutions to the problem, we extend these planners by using a previously suggested greedy algorithm to choose a subset of plans of higher diversity (Katz and Sohrabi 2020). For bQbD-k, we show that the decision problem that corresponds to the second step is NP-complete and suggest using previously proposed integer linear programming formulation. As the formulation was not previously formally described, we describe it formally and prove that it provides us with a solution to bQbD-k. For bQoptD-k, as the optimization problem in second step is NP-hard, we propose using a novel mixed integer linear program to solve it. We formally describe the program and prove that a solution to the program can be used for solving bQoptD-k. Our approach is the first one that is able to provide solutions to the problem.

Finally, we perform an empirical evaluation of our proposed planners. For bQbD-k, we show to favorably compare to the existing planners. For bQ-k, as no previous planners exist, we test the quality of our solution by comparing it to the quality of the optimal solution, obtained by our proposed planner for bQoptD-k, where no previous planners exist either. We show that the greedy algorithm works very well on tested domains, producing results close to the optimum. Our novel contributions, thus, include (i) the introduction of the new quality metric that allows us to connect between the cost of plans and quality of a set of plans, (ii) a concrete algorithmic scheme that uses top quality planners for the first step, finding a set of plans of bounded cost, (iii) computational complexity investigation of the second step, choosing a proper subset from the found set, for various computational problems, and (iv) introduction of the new mixed integer linear program for the resulting optimization problem.

## Preliminaries and Related Work

In this work we follow the notation of Katz and Sohrabi (2020). A $\text{SAS}^+$ *planning task* (Bäckström and Nebel 1995) is given by a tuple $\langle \mathcal{V}, \mathcal{A}, s_0, s_* \rangle$, with $\mathcal{V}$ being a set of *state variables* and $\mathcal{A}$ being a finite set of *actions*. Each state variable $v \in \mathcal{V}$ has a finite domain $dom(v)$ of values. A pair $\langle v, \vartheta \rangle$ with $v \in \mathcal{V}$ and $\vartheta \in dom(v)$ is called a *fact*. A (partial) assignment to $\mathcal{V}$ is called a *(partial) state*. Often it is convenient to view partial state $p$ as a set of facts with $\langle v, \vartheta \rangle \in p$ if and only if $p[v] = \vartheta$. Partial state $p$ is *consistent* with state $s$ if $p \subseteq s$. We denote the set of states of a planning task by $\mathcal{S}$. $s_0$ is the *initial state*, and the partial state $s_*$ is the *goal*. Each *action* $a$ is a pair $\langle pre(a), eff(a) \rangle$ of partial states called *preconditions* and *effects*. An *action cost* is a mapping $C : \mathcal{A} \to \mathbb{R}^{0+}$. An action $a$ is applicable in a state $s \in \mathcal{S}$ if and only if $pre(a)$ is consistent with $s$. Applying $a$ changes the value of $v$ to $eff(a)[v]$, if defined. The resulting state is denoted by $s[\![a]\!]$. An action sequence $\pi = \langle a_1, \ldots, a_k \rangle$ is applicable in $s$ if there exist states $s_0, \cdots, s_k$ such that (i) $s_0 = s$, and (ii) for each $1 \le i \le k$, $a_i$ is applicable in $s_{i-1}$ and $s_i = s_{i-1}[\![a_i]\!]$. We denote the state $s_k$ by $s[\![\pi]\!]$. $\pi$ is a plan iff $\pi$ is applicable in $s_0$ and $s_*$ is consistent with $s_0[\![\pi]\!]$. We denote by $\mathcal{P}(\Pi)$ (or just $\mathcal{P}$ when the task is clear from the context) the set of all plans of $\Pi$. The cost of a plan $\pi$, denoted by $C(\pi)$ is the summed cost of the actions in the plan.

In regard to reasoning about sets of plans rather than individual plans, there are two main measures defined on sets of plans, *quality* and *diversity*. Previous work has introduced one definition of quality, mirroring the International Planning Competition (IPC) quality metric for individual plans (Katz and Sohrabi 2020).

**Definition 1** *Let $P$ be the set of known plans of $\Pi$ and let $P' \subseteq P$ be a subset of plans. The relative quality of $P'$ with respect to $P$ is defined as*

$$Q_P(P') := \frac{1}{|P'|} \times \sum_{i=1}^{|P'|} \frac{C(\pi_i)}{C(\pi'_i)},$$

*where $\pi_1, \ldots, \pi_{|P'|}$ and $\pi'_1, \ldots, \pi'_{|P'|}$ are plans in $P$ and $P'$, respectively, sorted in ascending order of their costs.*

The relative quality of a set of plans is always between $0$ and $1$, being $1$ if and only if there is no plan in $P \setminus P'$ that is cheaper than any plan in $P'$.

Switching now our attention to diversity metrics, pairwise plan distance is defined by $\delta(\pi, \pi') = 1 - sim(\pi, \pi')$, where sim is a similarity measure, a value between $0$ (unrelated) and $1$ (equivalent). The diversity of a set of plans, $D(P)$, $P \subseteq \mathcal{P}$ is then defined as some aggregation (e.g., min or average) of the pairwise distance within the set $P$. In this work, we focus on one of the most popular similarity measures, *stability* (Fox et al. 2006; Coman and Muñoz-Avila 2011). Stability similarity measures the ratio of the number of actions that appear on both plans to the total number of actions on these plans, referring to plans as action multi-sets (sets with repetitions). Given two plans $\pi, \pi'$, it is defined as $sim_{\text{stability}}(\pi, \pi') = |A(\pi) \cap A(\pi')|/|A(\pi) \cup A(\pi')|$, where $A(\pi)$ is the multi-set of actions in $\pi$. In what follows,

by $D_{ma}$ we denote the diversity metric computed as minimum over the pairwise plan distance under stability similarity, the diversity metric implemented by multiple existing diverse planners (Nguyen et al. 2012; Coman and Muñoz-Avila 2011; Vadlamudi and Kambhampati 2016).

There is a variety of computational problems that fall under the umbrella of diverse planning. In our work, focusing on bounded quality problems, we follow the recently introduced taxonomy (Katz and Sohrabi 2020).

**Definition 2 (Diverse planning solution)** *Let $\Pi$ be a planning task and $\mathcal{P}$ be the set of all plans for $\Pi$. Given a natural number $k$, $P \subseteq \mathcal{P}$ is a $k$-diverse planning solution if $|P| = k$ or $P = \mathcal{P}$ if $|\mathcal{P}| < k$.*

**Definition 3 (Quality-bounded solution)** *Let $\Pi$ be a planning task, $Q$ be some quality metric, $c$ be some bound, and $\mathcal{P}$ be the set of all $\Pi$'s plans. Given a natural number $k$, $P \subseteq \mathcal{P}$ is a $c$-quality-bounded $k$-diverse planning solution if it is a $k$-diverse planning solution and $Q(P) \geq c$.*

Bounded Quality Diverse Planning computational problem is defined as follows.

$$\text{bQ-k : Given } k \text{ and } c, \text{ find a } c\text{-quality-bounded}$$
$$k\text{-diverse planning solution.}$$

**Definition 4 (Diversity-bounded solution)** *Let $\Pi$ be a planning task, $D$ be some diversity metric, $b$ be some bound, and $\mathcal{P}$ be the set of all $\Pi$'s plans. Given a natural number $k$, $P \subseteq \mathcal{P}$ is a $b$-diversity-bounded $k$-diverse planning solution if it is a $k$-diverse planning solution and $D(P) \geq b$.*

Bounded Quality and Diversity Diverse Planning computational problem is defined as follows.

$$\text{bQbD-k : Given } k, b, \text{ and } c, \text{ find a } c\text{-quality-bounded}$$
$$\text{and } b\text{-diversity-bounded } k\text{-diverse planning solution.}$$

Note that the definition above generalizes the previously suggested search problem described in Equation 1 below and implemented for the diversity metric $D_{ma}$ by Vadlamudi and Kambhampati (2016).

$$\text{cCOSTdDISTANTkSET : find } P \text{ with } P \subseteq \mathcal{P},$$
$$|P| = k, \min_{\pi, \pi' \in P} \delta(\pi, \pi') \geq d, C(\pi) \leq c \; \forall \pi \in P. \quad (1)$$

Finally, Optimal Diversity Bounded Quality Diverse Planning optimization problem is defined as follows.

$$\text{bQoptD-k : Given } k \text{ and } c, \text{ find a diversity-optimal}$$
$$\text{among } c\text{-quality-bounded } k\text{-diverse planning solutions.}$$

## Bounded Quality in Diverse Planning

As stated above, in this work, we focus on the three computational problems in diverse planning taxonomy of Katz and Sohrabi (2020) that deal with bounded quality, bQ-k, bQbD-k, and bQoptD-k. Our proposed solutions to these three problems all have in common the first step - finding a set of plans of bounded quality. While existing planners for bounded quality diverse planning took the same

approach (Vadlamudi and Kambhampati 2016), they used planners for the top-k planning problem (Riabov, Sohrabi, and Udrea 2014). Specifically, A*AA/A*AM apply the $m$-$A^*$ algorithm (Flerova, Marinescu, and Dechter 2016), while DFAA/DFAM apply the well-known branch-and-bound algorithm. We suggest using a different approach, generating plans with a planner for a recently proposed unordered top-quality problem (Katz, Sohrabi, and Udrea 2020). Switching to top-quality allows to ensure that *all* plans of bounded cost are found. Unordered top-quality allows to disregard plans that are reorderings of the found plans. For some diversity metrics, this is highly beneficial, with pairwise diversity between a plan and its reordering might be very low. Further, ignoring plan orders reduces the computational effort required for finding all plans of bounded cost.

The second step, after finding a set of plans of bounded quality, is different for the different computational problems that we consider in this work. For bQ-k, although any set of $k$ plans is a solution, we strive to obtain solutions of higher diversity. Therefore, we apply a greedy algorithm that iteratively increases the set of plans by adding at each step the candidate plan that increases the overall diversity score the most, the same algorithm that was used for satisficing diverse planning (Katz and Sohrabi 2020). For bQbD-k and bQoptD-k, we cast the problem of finding the subset of plans as (mixed) integer linear programs. We describe these programs in detail in what follows. We start with the discussion of the quality metric that we consider in this work.

### Quality Metric

While we consider the planners DFAA/DFAM and A*AA/A*AM to be solving the Bounded Quality and Diversity Diverse Planning problem as defined by Katz and Sohrabi (2020), the quality metric they maximize is not obvious. These planners consider the set $P$ of plans of cost smaller or equal than a given absolute bound value $c$, or $\max_{\pi \in P} C(\pi) \leq c$. Alternatively, the criterion can be expressed via $Q_a(P) \geq c'$ for the quality measure

$$Q_a(P) = \frac{c^*}{\max_{\pi \in P} C(\pi)} \quad (2)$$

where $c^*$ is the task's optimal plan cost and $c' = \frac{c^*}{c} \in [0, 1]$. Thus, the planners above solve the bQbD-k problem for the quality metric $Q_a$ as in Eq. 2. Note that this quality measure is different from the measure described in Definition 1, as introduced by Katz and Sohrabi (2020), where the quality of the set of plans is affected by the costs of all plans, not only most expensive ones. The above proposed quality metric makes it possible to connect the quality metric to the cost bound.

### Bounded Quality Planning

Let us consider now the (unordered) top quality planners (Katz, Sohrabi, and Udrea 2020), that, given a multiplier $q_m \geq 1$, return the set of all plans $P$ such that $\forall \pi \in P$ we have $C(\pi) \leq q_m \times c^*$, or a subset thereof with a single representative for plans that differ only in the order of their actions for the unordered case. For such sets, we have

$Q_a(P) \geq \frac{1}{q_m}$, and therefore these planners can be used to derive solutions of bounded quality according to the quality metric $Q_a$. Furthermore, top quality planners produce a set of plans that is a super-set of the sets of plans that constitute solutions to all three computational problems of interest, bQ-k, bQbD-k, and bQoptD-k. Therefore, in what follows, we will focus on finding subsets of plans out of a given set of plans, according to the relevant solution definition for the corresponding computational problem.

Focusing first on bQ-k, while any subset of required size of the set of plans returned by top quality planners is a solution to bQ-k, different subsets can vary significantly in their diversity measure score. Since bQ-k does not pose any restrictions on these subsets beyond the desired size, one possible way of coming up with subsets of high diversity is to employ the same greedy selection algorithm that was used for Satisficing Diverse Planning (Katz and Sohrabi 2020). The algorithm iteratively constructs a set of plans by greedily adding a plan that will contribute the most to already added plans.

## Bounding Diversity

Switching now our attention to bQbD-k, first, note that for a set of plans and a number $k$, the decision problem of whether there exists a subset of bounded diversity of size $k$ is NP-complete. The membership in NP is trivial. We show the hardness by a polynomial reduction from the clique problem (Garey and Johnson 1979).

For a graph $G = (V, E)$, let $P_G = \{\pi_v \mid v \in V\}$ be a collection of plans. For a pair of plans $\pi_u, \pi_v \in P$

$$d(\pi_u, \pi_v) = \begin{cases} d, & (u, v) \in E, \\ 0, & \text{otherwise.} \end{cases}$$

**Theorem 1** *Given a number $k$ and diversity bound $d > 0$ for diversity metrics minimal pairwise diversity, if there is a subset of plans $P_G$ of bounded by $d$ diversity of size at least $k$, then there is a clique in $G$ of the same size.*

**Proof:** Let $P \subseteq P_G$ be a subset of plans such that $|P| \geq k$ and $D(P) \geq d$. Then for all $\pi_u, \pi_v \in P$ we have $d(\pi_u, \pi_v) \geq d$ and therefore $d(\pi_u, \pi_v) > 0$. Thus, it must be the case that $(u, v) \in E$ for all $\pi_u, \pi_v \in P$ and thus the set $V' = \{v \mid \pi_v \in P\}$ is a clique, of size $|V'| = |P| \geq k$. ∎

Next, we describe the mixed integer linear program that is used for finding a subset of plans of bounded diversity. While the program is not novel,[1] its description was not presented in the literature. Here, we describe the program in detail. Given a set of plans $P$ and a bound on the diversity $d$, the variables are as follows.

- A binary variable $v_\pi$ per plan $\pi \in P$, describing whether the plan is selected for the subset.

The constraints are as follows.

(i) $\forall \pi, \pi' \in P$, s.t. $d(\pi, \pi') < d : v_\pi + v_{\pi'} \leq 1$, stating that if the pairwise diversity of $\pi$ and $\pi'$ is below $d$,

---

[1]The mixed integer linear program was previously used for bounded diversity diverse planning (Katz and Sohrabi 2020)

then at most one of these plans can be selected for the subset, and

(ii) $\sum_{\pi \in P} v_\pi \geq k$, forcing the size of the subset be at least $k$.

The objective of the program is to minimize $\sum_{\pi \in P} v_\pi$. In words, the program encodes a subset selection and restrict the selected subset to not have pairs of plans with diversity outside of the provided bound. In what follows, we prove that the program can be used for devising solutions for bQbD-k.

**Theorem 2** *For a planning task $\Pi$ with a set of all plans of bounded quality $P$ such that $|P| \geq k$, and a bound $d$, the binary program finds a subset of size $k$ with the bounded by $d$ diversity score, for diversity metrics maximizing minimal pairwise diversity, if such subset exists. Otherwise, the program is infeasible.*

**Proof:** We first show that a solution to bQbD-k corresponds to a feasible assignment. Let $P' \subset P$ be a solution to bQbD-k for the bound $d$, with $|P'| = k$. Then, let $\overline{v}$ assign 1 to plans $\pi \in P'$ and 0 otherwise. Since $|P'| = k$, constraint (ii) holds. For $\pi, \pi' \in P$, if either of the plans is not in $P'$, then $\overline{v_\pi} + \overline{v_{\pi'}} \leq 1$. If both plans are in $P'$, then $d(\pi, \pi') \geq d$. Thus, constraint set (i) holds for $\overline{v}$ and the program is feasible.

Now, let $\overline{v}$ be some feasible solution and let $P' = \{\pi \in P \mid \overline{v_\pi} = 1\}$ be the corresponding subset of $P$. Then, (a) from constraint (ii) we have $|P'| \geq k$, and (b) for all $\pi, \pi' \in P'$, since $\overline{v_\pi} + \overline{v_{\pi'}} = 2$, we know that the corresponding constraint is not in the constraint set (i), and therefore $d(\pi, \pi') \geq d$. Therefore, $P'$ (or any of its subset of size $k$) is a solution to bQbD-k. ∎

We now switch our attention to the next computational problem, bQoptD-k.

## Optimizing Diversity

Due to the NP-completeness of the decision problem of selecting a subset of plans of bounded diversity, the corresponding optimization problem is NP-hard. To solve it efficiently, we encode it in mixed integer linear programming. We present a novel mixed integer linear program that we use for finding a subset of size $k$ that optimizes the diversity metric. Given a set of plans $P$, we define the variables as follows.

- A binary variable $v_\pi$ per plan $\pi \in P$, describing whether the plan is selected for the subset, and

- a single continuous variable $d$ for bounding the pairwise diversity.

The constraints are as follows.

(i) $\sum_{\pi \in P} v_\pi = k$, stating that the size of the subset is exactly $k$, and

(ii) $\forall \pi, \pi' \in P : d + v_\pi + v_{\pi'} \leq d(\pi, \pi') + 2$, stating that $d$ is bounded by the diversity of each chosen pair, if the pair is chosen.

| Coverage | $q_m=1.00$ | | | | $q_m=1.05$ | | | | $q_m=1.10$ | | | | $q_m=1.20$ | | | |
|---|---|---|---|---|---|---|---|---|---|---|---|---|---|---|---|---|
| | DFA | A*A | FI | Sym | DFA | A*A | FI | Sym | DFA | A*A | FI | Sym | DFA | A*A | FI | Sym |
| airport (28) | 0 | 7 | **18** | 7 | 0 | 7 | **18** | 7 | 0 | 7 | **17** | 7 | 0 | 7 | **17** | 7 |
| barman11 (8) | 0 | 0 | 4 | **8** | 0 | 0 | **5** | 4 | 0 | 0 | **5** | 4 | 0 | 0 | **5** | 4 |
| barman14 (4) | 0 | 0 | 3 | **4** | 0 | 0 | 3 | 3 | 0 | 0 | 3 | 3 | 0 | 0 | 2 | **4** |
| blocks (30) | 12 | 21 | 18 | **28** | 11 | 20 | 19 | **29** | 18 | 20 | 18 | **28** | 20 | 20 | 17 | **22** |
| childsnack14 (6) | 0 | 0 | 0 | **1** | 0 | 0 | 0 | **1** | 0 | 0 | 0 | **1** | 0 | 0 | 0 | **1** |
| data-ntwrk18 (13) | 9 | 7 | 9 | **11** | **10** | 7 | **10** | 10 | 9 | 7 | **10** | 10 | 9 | 7 | **10** | 10 |
| depot (12) | 0 | 1 | **3** | **3** | 0 | 1 | **3** | **3** | 2 | 1 | **3** | **3** | 2 | 1 | **3** | **3** |
| driverlog (14) | 3 | 6 | **10** | 9 | 2 | 6 | **10** | 9 | 6 | 2 | **10** | 9 | 8 | 2 | **10** | 9 |
| elevators08 (24) | 4 | 2 | **7** | 0 | 4 | 2 | **7** | 0 | 4 | 2 | **5** | 0 | 2 | 2 | **6** | 0 |
| elevators11 (18) | 3 | 1 | **5** | 0 | 2 | 1 | **5** | 0 | 2 | 1 | **5** | 0 | 2 | 1 | **5** | 0 |
| floortile11 (14) | 2 | 0 | 2 | **5** | 2 | 0 | 2 | **5** | 2 | 0 | 2 | **5** | 2 | 0 | 2 | **5** |
| floortile14 (20) | 0 | 0 | 0 | **8** | 0 | 0 | 0 | **8** | 0 | 0 | 0 | **7** | 0 | 0 | 0 | **8** |
| freecell (22) | 6 | 6 | 7 | **21** | 6 | 6 | 7 | **21** | 6 | 6 | 7 | **21** | 8 | 6 | 7 | **18** |
| ged14 (19) | 12 | 13 | 12 | **15** | 12 | 13 | 12 | **15** | 12 | 13 | 12 | **15** | 12 | 13 | 12 | **15** |
| grid (2) | **2** | 1 | 1 | **2** | **2** | 1 | 1 | **2** | **2** | 1 | 1 | **2** | **2** | 1 | 1 | **2** |
| gripper (20) | 3 | 2 | 6 | **15** | 3 | 2 | 6 | **15** | 4 | 2 | 6 | **15** | 8 | 2 | 6 | **15** |
| hiking14 (19) | 3 | 2 | **7** | 7 | 3 | 2 | **8** | 6 | 2 | 1 | **6** | 6 | 3 | 1 | **8** | 5 |
| logistics00 (20) | 0 | 3 | **15** | 5 | 0 | 3 | **12** | 4 | 0 | 3 | **10** | 3 | 1 | 0 | **6** | 3 |
| logistics98 (6) | 1 | 0 | **5** | 2 | 1 | 0 | **5** | 2 | 3 | 0 | **5** | 2 | 3 | 0 | **5** | 2 |
| miconic (143) | **121** | 70 | 53 | 79 | **121** | 67 | 54 | 70 | **122** | 68 | 53 | 63 | **118** | 67 | 53 | 57 |
| movie (30) | **30** | **30** | **30** | 2 | **30** | **30** | **30** | 2 | **30** | **30** | **30** | 2 | **30** | **30** | **30** | 0 |
| mprime (24) | 6 | **21** | 20 | 17 | 6 | **21** | 20 | 17 | 6 | **21** | 20 | 17 | 15 | **21** | 19 | 16 |
| mystery (17) | 1 | **16** | **16** | 11 | 1 | **16** | **16** | 12 | 1 | **16** | 15 | 12 | 10 | **16** | 15 | 11 |
| nomystery11 (17) | 4 | 11 | 11 | **13** | 7 | 7 | 10 | **13** | 10 | 6 | 9 | **12** | 11 | 4 | 9 | **13** |
| openstacks08 (30) | **26** | 4 | 17 | 21 | **26** | 4 | 17 | 21 | **26** | 4 | 17 | 21 | **26** | 4 | 17 | 21 |
| openstacks11 (20) | **18** | 1 | 12 | 16 | **18** | 1 | 12 | 15 | **18** | 1 | 12 | 15 | **18** | 1 | 12 | 16 |
| openstacks14 (18) | **5** | 0 | 2 | 1 | **5** | 0 | 2 | 2 | **5** | 0 | 2 | 1 | **6** | 0 | 2 | 1 |
| openstacks (17) | **5** | 0 | **5** | 0 | **5** | 0 | **5** | 0 | **5** | 0 | **5** | 0 | 0 | 0 | **5** | 0 |
| organic-s18 (7) | 6 | **7** | **7** | **7** | 6 | **7** | **7** | **7** | 6 | **7** | **7** | **7** | 6 | **7** | **7** | **7** |
| organic-ssp18 (15) | 11 | **14** | **14** | 12 | 11 | **14** | **14** | 13 | 11 | 14 | **15** | 12 | 11 | 14 | **15** | 11 |
| parcprinter08 (30) | 0 | 6 | **15** | 6 | 1 | 4 | **13** | 6 | 1 | 4 | **13** | 6 | 1 | 4 | **13** | 6 |
| parcprinter11 (20) | 0 | 3 | **11** | 3 | 1 | 1 | **9** | 3 | 1 | 1 | **9** | 3 | 1 | 1 | **8** | 3 |
| parking11 (3) | **2** | 1 | 0 | 0 | **1** | **1** | **1** | 0 | **1** | **1** | **1** | 0 | **2** | 1 | 1 | 0 |
| parking14 (4) | **2** | **2** | 0 | 0 | **2** | **2** | 0 | 0 | **1** | **1** | 0 | 0 | **2** | 1 | 0 | 0 |
| pegsol08 (30) | 9 | 21 | 28 | **29** | 9 | 21 | 27 | **29** | 9 | 21 | 28 | **29** | 20 | 20 | 28 | **29** |
| pegsol11 (20) | 9 | 9 | 18 | **19** | 9 | 9 | 17 | **19** | 11 | 8 | 18 | **19** | 16 | 6 | 17 | **19** |
| pipes-notank (22) | **14** | 11 | **14** | 12 | **14** | 11 | **14** | 12 | **14** | 11 | **14** | 12 | **16** | 11 | 14 | 10 |
| pipes-tank (18) | 7 | 6 | **13** | 8 | 7 | 6 | **13** | 9 | 5 | 6 | **14** | 7 | 7 | 6 | **13** | 6 |
| psr-small (50) | 4 | 31 | **46** | 44 | 4 | 30 | 42 | **45** | 5 | 29 | 40 | **43** | 13 | 26 | 38 | **42** |
| rovers (12) | 3 | 4 | **6** | **6** | 2 | 4 | **6** | **6** | 4 | 4 | 5 | **6** | **6** | 4 | 5 | **6** |
| satellite (15) | 7 | 5 | 7 | **11** | 6 | 5 | 7 | **11** | 6 | 5 | 7 | **9** | 6 | 5 | **7** | **7** |
| scanalyzer08 (19) | 12 | **13** | 12 | 12 | 7 | **13** | **13** | 12 | 6 | **13** | **13** | 10 | 8 | **13** | **13** | 10 |
| scanalyzer11 (15) | **11** | 10 | 10 | 9 | 6 | **10** | **10** | 9 | 5 | **10** | **10** | 7 | 7 | **10** | **10** | 7 |
| snake18 (11) | **5** | 4 | 3 | 3 | **5** | 4 | 3 | 3 | **6** | 4 | 3 | 3 | **7** | 4 | 3 | 3 |
| sokoban08 (30) | 5 | **6** | 0 | 0 | 5 | **6** | 0 | 0 | 3 | **6** | 0 | 0 | 3 | **6** | 0 | 0 |
| sokoban11 (20) | **3** | **3** | 0 | 0 | **3** | **3** | 0 | 0 | 1 | **3** | 0 | 0 | 0 | **3** | 0 | 0 |
| spider18 (11) | 5 | **7** | 6 | 2 | 6 | **7** | 4 | 3 | **8** | 7 | 5 | 3 | **8** | 7 | 5 | 3 |
| storage (18) | 8 | 11 | **16** | 14 | 8 | 11 | **16** | 14 | 7 | 11 | **16** | 14 | 11 | 11 | **16** | 13 |
| termes18 (16) | 0 | 0 | 0 | **10** | 0 | 0 | 0 | **9** | 0 | 0 | 0 | **10** | 0 | 0 | 0 | **10** |
| tetris14 (9) | 1 | 1 | 3 | **5** | 1 | 1 | 3 | **5** | 2 | 1 | 3 | **5** | 4 | 1 | 3 | **5** |
| tidybot11 (16) | 1 | 5 | 6 | **10** | 4 | 3 | 4 | **10** | **12** | 2 | 3 | 9 | **12** | 1 | 3 | 9 |
| tidybot14 (10) | **4** | 2 | 0 | 3 | **6** | 2 | 0 | 2 | **6** | 2 | 0 | 3 | **7** | 2 | 0 | 3 |
| tpp (11) | 0 | 3 | **6** | 4 | 0 | 3 | **6** | 4 | 0 | 3 | **6** | 4 | 0 | 3 | **6** | 4 |
| transport08 (12) | 4 | 5 | 8 | **11** | 5 | 5 | **8** | 4 | 6 | 5 | **8** | 4 | 6 | 5 | **8** | 4 |
| transport11 (8) | 2 | 1 | 3 | **7** | 2 | 1 | **3** | **3** | 2 | 1 | **3** | **3** | 2 | 1 | **3** | **3** |
| transport14 (7) | 1 | 0 | 2 | **5** | 0 | 0 | 2 | **3** | 0 | 0 | **2** | **2** | 1 | 0 | **2** | **2** |
| trucks (12) | 6 | 3 | **7** | 5 | 5 | 3 | **7** | 6 | 6 | 3 | **7** | 5 | **7** | 3 | **7** | 5 |
| visitall11 (12) | 7 | 9 | 9 | **11** | 8 | 9 | 9 | **10** | **9** | 9 | 9 | 7 | **9** | 9 | 9 | 5 |
| visitall14 (6) | **6** | 3 | 3 | 5 | **6** | 3 | 3 | 2 | **6** | 3 | 3 | 0 | **6** | 3 | 3 | 0 |
| woodwork08 (28) | 8 | 7 | 10 | **22** | 10 | 6 | 10 | **22** | 14 | 6 | 10 | **22** | 12 | 6 | 10 | **22** |
| woodwork11 (20) | 7 | 2 | 5 | **16** | 8 | 2 | 4 | **16** | 9 | 2 | 5 | **16** | 7 | 2 | 5 | **16** |
| zenotravel (13) | 7 | **9** | 8 | **9** | 7 | 9 | 8 | **10** | 6 | 9 | 8 | **10** | 8 | 9 | 8 | **10** |
| Sum other(27) | 0 | 0 | 0 | 0 | 0 | 0 | 0 | 0 | 0 | 0 | 0 | 0 | 0 | 0 | 0 | 0 |
| Sum (1192) | 453 | 449 | 594 | **631** | 452 | 433 | 582 | **603** | 484 | 424 | 573 | **574** | 548 | 411 | **564** | 548 |

Table 1: Domain-wise coverage comparison of FI-bQbD and Sym-bQbD to DFAM and A*AM, for $k=5$, diversity bound $0.15$, and four quality bounds.

| | $q_m\!=\!1.00$ | | | | $q_m\!=\!1.05$ | | | | $q_m\!=\!1.10$ | | | | $q_m\!=\!1.20$ | | | |
|---|---|---|---|---|---|---|---|---|---|---|---|---|---|---|---|---|
| k | DFA | A*A | FI | Sym | DFA | A*A | FI | Sym | DFA | A*A | FI | Sym | DFA | A*A | FI | Sym |
| 10 | 390 | 373 | 530 | **594** | 389 | 354 | 511 | **559** | 426 | 338 | 494 | **535** | **505** | 318 | 480 | 499 |
| 100 | 189 | 238 | 433 | **491** | 194 | 216 | 403 | **448** | 229 | 185 | 365 | **396** | 291 | 123 | 302 | **352** |
| 1000 | 79 | 238 | 380 | **435** | 82 | 216 | 345 | **386** | 105 | 185 | 287 | **321** | 124 | 123 | 203 | **257** |

Table 2: The overall coverage comparison of FI-bQbD and Sym-bQbD to DFAM and A*AM, for diversity bound 0.15, four quality bounds, and various $k$ values.

The objective of the program is then to maximize $d$. In words, as in the previous case, the program encodes a subset selection, but in this case all subsets of size $k$ correspond to valid assignments. We additionally have a continuous variable $d$ that is bounded by the diversity score of the selected subset. In the case of diversity metrics that correspond to minimal pairwise diversity, this would mean to require the variable $d$ to be bounded by the diversity of each selected pair. In other words, if a pair of plans is selected, then $d$ should be no greater than their diversity score. If a pair is not selected, there is no such restriction, but since there is a natural upper bound of 1 on the overall diversity, $d$ can be required to be upper bounded by any value that is larger or equal to 1. If at least one of $v_\pi$, $v_{\pi'}$ gets 0 assigned to it, the constraint $d + v_\pi + v_{\pi'} \leq d(\pi, \pi') + 2$ is then satisfied. Therefore, the constraint is valid whether the variables $v_\pi$ and $v_{\pi'}$ are assigned 0 or 1.

In what follows, we prove that the program can be used for devising solutions for bQbD-k.

**Theorem 3** *For a planning task $\Pi$ with a set of all plans of bounded quality $P$ such that $|P| \geq k$, the mixed integer program finds a subset of size $k$ with the optimal diversity score, for diversity metrics maximizing minimal pairwise diversity.*

**Proof:** Let $\overline{v}, \overline{d}$ be a feasible assignment to the variables of the mixed integer program and let $P' = \{\pi \in P \mid \overline{v_\pi} = 1\}$ be the corresponding subset of $P$. Then, from the constraint set (i) we have $|P'| = k$ and from constraint set (ii) we have $d \leq d(\pi, \pi')$ for all $\pi, \pi' \in P'$. Further, for a plan $\pi \in P \setminus P'$ and a plan $\pi' \in P'$ we have $d \leq 1 + d(\pi, \pi')$, which does not pose additional constraint on the values of $d$ since all pairwise distances $d(\pi, \pi')$ are upper-bounded by 1. Similarly, for $\pi, \pi' \in P \setminus P'$, we have $d \leq 2 + d(\pi, \pi')$, which also does not pose additional constraint on the values of $d$. Therefore we have $\overline{d} \leq d(\pi, \pi')$ for all $\pi, \pi' \in P'$ and maximizing $d$ without changing $\overline{v}$ would lead to $\overline{d} = \min_{\pi, \pi' \in P'} d(\pi, \pi')$. Thus, the linear program finds a subset of size $k$ with maximum minimal pairwise diversity. ∎

## Experimental Evaluation

To empirically evaluate the feasibility of our suggested approach, we have implemented our diverse planners on top of the Diversity Score Computation component (Katz and Sohrabi 2019), using CPLEX v12.8.0 for solving the mixed integer linear programs. The code is available at https://github.com/IBM/diversescore. The experiments were performed on Intel(R) Xeon(R) CPU E7-8837 @2.67GHz machines, with the time and memory limit of 30min and 2GB, respectively. The benchmark set consists of all STRIPS benchmarks from optimal tracks of International Planning Competitions (IPC) 1998-2018, a total of 1797 tasks in 64 domains. For Bounded Quality and Diversity Diverse Planning (bQbD-k), we compare to the existing planners for that computational problem DFAM and A*AM (Vadlamudi and Kambhampati 2016). Since these planners are implemented for the diversity metric $D_{ma}$, we focus our experimental evaluation on $D_{ma}$, although our approach works with any metric. Further, since these planners require an absolute bound on the solution cost to be provided as a parameter, we further restrict the benchmark set to tasks where optimal costs could be found with a state-of-the-art cost-optimal planner. For that, we used the 17 single planners from the portfolio of Delfi1 (Katz et al. 2018a). As a result, for the bQbD-k computational problem, the benchmark set consists of 1192 tasks.

As a first step, we generate a set of plans of bounded quality. Focusing on $D_{ma}$ allows us to use unordered top-quality planners (Katz, Sohrabi, and Udrea 2020) to derive all plans (modulo reorderings) of bounded cost. This is due to the fact that two plans that differ only in the order of their actions would produce pairwise diversity of 0 and thus any set of plans $P$ that includes two such plans would get $D_{ma}(P) = 0$. For other diversity metrics we might need to produce the set of all plans of bounded cost (Katz, Sohrabi, and Udrea 2020). Note that some top-k planners, such as $K^*$-based (Katz et al. 2018b) and symbolic search based (Speck, Mattmüller, and Nebel 2020) can be easily adapted to produce solutions for top-quality planning. Further, these two planners can be rather naively adapted to produce unordered toq-quality solutions, by performing a duplicate check and skipping plans for which a reordering was previously found. In our experiments, we have performed the first step with each of these three planners, namely *FI*, $K^*$, and *Sym*. We run these planners with a 29min time bound, to allow at least one minute for the second step. In all cases, the overall time bound for both steps is 30min. Further, to avoid generating a larger amount of plans, the overall bound on the number of generated plans for the first step is set to 10000. As a second step, we select a subset of plans according to the computational problem of interest. For bQ-k, we use the greedy approach suggested by Katz and Sohrabi (2020). For the bQbD-k and bQoptD-k computational problems, we solve a mixed-integer linear program, as described in the previous section. This results in three configurations for each computational problem of interest. For space reasons, in what follows, we focus on the two best performing ones, *FI* and *Sym*.

Table 1 presents a domain-wise comparison of our planners, FI-bQbD and Sym-bQbD to the existing planners DFAM and A*AM (Vadlamudi and Kambhampati 2016), for

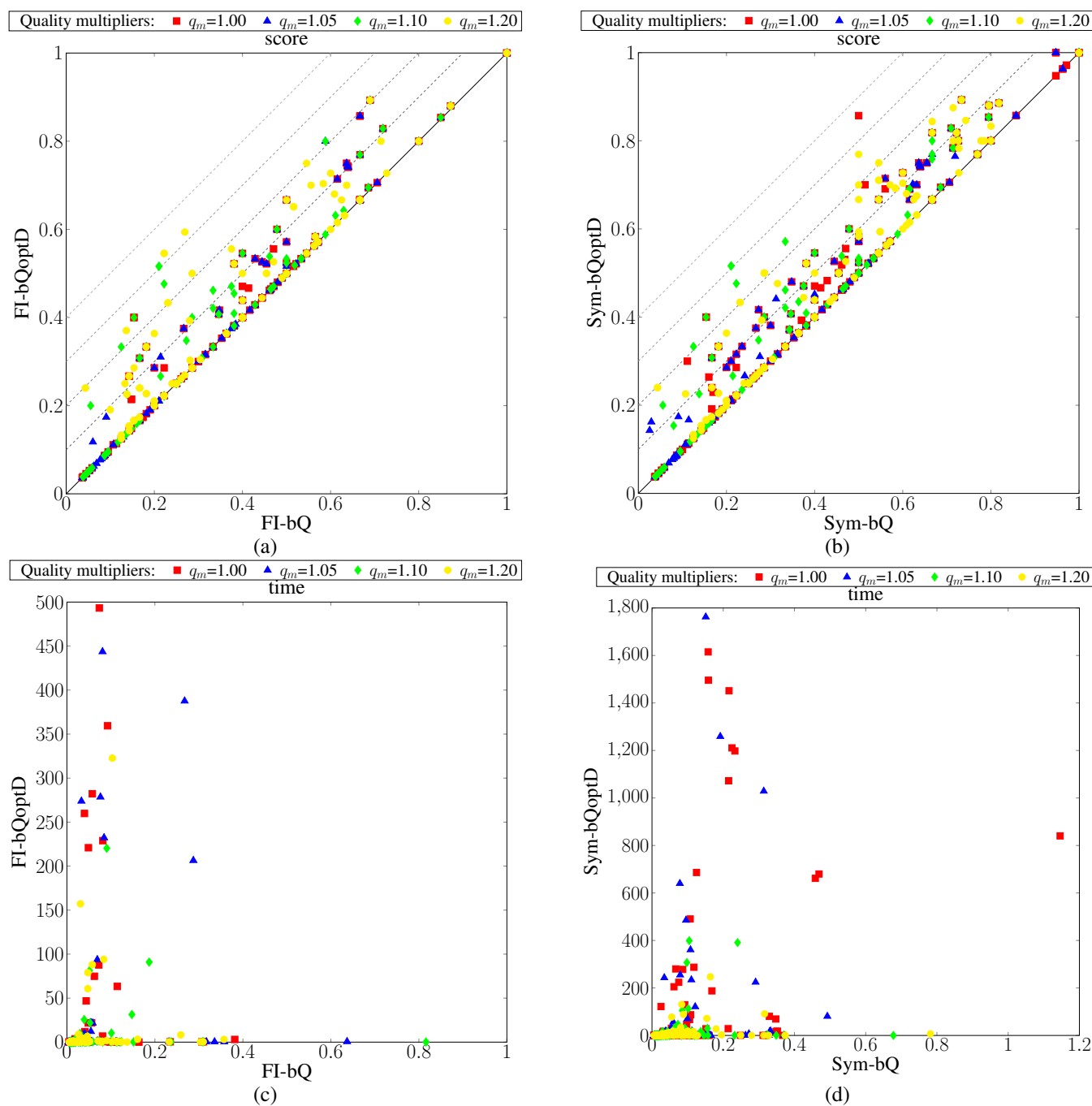

Figure 1: Comparison of the greedy and the optimal approaches to subset selection for $k = 5$. FI-bQ vs. FI-bQoptD: (a) diversity score and (c) score computation time. Sym-bQ vs. Sym-bQoptD: (b) diversity score and (d) score computation time.

$k = 5$. We use the diversity bound of $0.15$ and experiment with four quality bounds, defined by multipliers of the optimal plan cost, from $q_m = 1.0$ (optimal plans only), to $q_m = 1.05$, $q_m = 1.10$, and to $q_m = 1.20$ (up to 120% of the optimal plan cost). The results for these four quality bound multipliers are depicted in the four parts of the table. Each part presents the coverage value for the four plan-

ners. A planner gets a coverage of 1 on a planning task if it was able to either find a solution of size $k$ or prove that no such solution exists. Otherwise, the planner gets coverage 0. The coverage of a domain is a sum over coverages of all tasks in the domain. Best results are highlighted in bold. While both FI-bQbD and Sym-bQbD outperform the existing approaches in terms of overall coverage, it is worth

mentioning that for each of the approaches there are multiple domains where that approach exhibit superior behavior. To show how these planners scale with larger values of $k$, Table 2 presents aggregated overall coverage for the values of $k = 10, 100, 1000$. Going deeper into the coverage results, note that DFAM does not prove unsolvability. A*AM, on the other hand, for large values of $k = 100$ and $k = 1000$, did not find any solutions, and all the instances reported in the table for A*AM and these values of $k$ correspond to unsolvable cases. For our suggested approach, both FI-bQbD and Sym-bQbD are able to cope with both, for any value of $k$. It is worth mentioning that the performance of DFAM often improves, sometimes significantly, with larger quality bounds. We conjecture that this increase is due to the nature of the branch-and-bound algorithm, that does not necessarily produce plans in the order of their costs.

Switching to bQ-k and bQoptD-k, in order to evaluate the quality of the solution obtained using the greedy algorithm, we compare the diversity metric score of the subset chosen by FI-bQ (respectively, Sym-bQ) to the best possible score, obtained with the FI-bQoptD (respectively, Sym-bQoptD) planner. Figure 1(a,b) depict the comparison for $k = 5$, for all four quality multipliers 1.0, 1.05, 1.1, and 1.2, for tasks where both planners were able to find a solution, and the first step produced at least $k + 1$ plans. Note that the greedy approach works surprisingly well. On these tasks, in most cases the greedy algorithm has reached the optimum (nodes on the diagonal): 80 out of 121, 90 out of 126, 71 out of 105, and 50 out of 105 for FI-bQ and 106 out of 176, 99 out of 162, 67 out of 118, and 63 out of 129 for Sym-bQ (for the four quality multipliers, respectively). When it hasn't reached the optimum, the scores are still mostly below the $y = x + 0.1$ line. There are only 21, 18, 17, and 26 tasks for FI-bQ and 26, 23, 20, and 30 tasks for Sym-bQ for the four quality multipliers 1.0, 1.05, 1.1, and 1.2, respectively above the $y = x + 0.1$ line, and only 21 tasks for FI-bQ and 16 tasks for Sym-bQ in total for all quality multipliers above the $y = x + 0.2$ line.

While our experiments show that the greedy approach often produces solutions of diversity close to optimum, the question remains how these algorithms compare in their run time. Figure 1 (c) and (d) present such run time comparison between the greedy and the optimal approaches. The greedy algorithm always finished in under 1.2 seconds, while solving mixed integer linear program takes significantly longer on these tasks, up to 500 seconds for FI-bQoptD and 1760 seconds for Sym-bQoptD.

Finally, note that an inherent limitation of our approach to solving bQoptD-k is that the first step must produce a solution to the (unordered) top quality planning problem. There is no such limitation when solving bQ-k. As a result, FI-bQ successfully solves bQ-k in 617, 617, 615, and 613 tasks for the four quality multipliers, while FI-bQoptD solves bQoptD-k in only 369, 333, 273, and 191 tasks. Similarly, Sym-bQ successfully solves bQ-k in 702, 675, 648, and 624 tasks for the four quality multipliers, while Sym-bQoptD solves bQoptD-k in only 379, 338, 262, and 196 tasks.

## Discussion and Future Work

In this work, we extend the portfolio of existing tools for various computational problems in diverse planning by introducing three new such tools. We follow the recently introduced taxonomy and, focusing on bQbD-k, map existing planners DFAA/DFAM and A*AA/A*AM to that problem. For that, we introduce a novel quality metric under which these planners can be considered to solve bQbD-k. The metric also allows us to use top quality planners as a basis for our proposed planners, for bQbD-k as well as for other computational problems, choosing a subset of plans from the solution for the top quality problem. We show that it is NP-complete to find a solution to bQbD-k, given the set of all plans of bounded cost and suggest using a previously proposed integer linear programming based approach, which is experimentally shown to favorably compete with existing planners. As the integer linear program was not previously detailed in the literature, we present it in detail and formally prove that it can be used for solving bQbD-k. Switching from bounding to optimizing diversity, we suggest a novel mixed integer linear program and formally prove that this program solves bQoptD-k. For another computational problem, bQ-k, we use an existing greedy approach of selecting a subset of plans, and empirically show that such a simple approach is able to often achieve the optimum in practice.

Our suggested approach is similar to the one of Vadlamudi and Kambhampati (2016), in that it is also separated into two steps: (i) finding a set of plans of bounded cost, and (ii) choosing a proper subset from the found set. There are two major differences. The first one is the stopping criteria for step (i): while Vadlamudi and Kambhampati (2016) iterate until enough plans are found or no more plans exist, and can stop before finding all plans of bounded cost, we are using an existing (unordered) top-quality planner as is, and therefore will produce the set of all plans of bounded cost in step (i). While it is possible to adapt the top quality planner that we used to terminate earlier, we decided not to do so, to allow for easily replacing the top quality planner with a different one. The second major difference is that instead of trying to construct a feasible solution during the execution of step (i), we perform step (ii) after the first step is finished, as a post-processing. Further, instead of implementing a dedicated algorithm, we cast the problem of choosing the proper subset as an integer linear program, allowing us to use existing solvers. Thus, our solution is highly modular, allowing us to easily replace the solvers when better ones become available.

While there has been significant progress in the field of diverse planning recently, there are still several interesting computational problems for which no planners currently exist. For example, in this work we show how to optimize diversity when the set of candidate plans is given. However, if the quality restriction is alleviated, it is not clear how to choose a set of maximal diversity. It is not even clear whether all plans must be considered while searching for such a set. Another possible problem of interest is finding a subset of optimal quality among the bounded diversity ones. Focusing on these planning problems is an interesting research direction.

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
