# OpenReview forum: "Bounding Quality in Diverse Planning"
_icaps-conference.org/ICAPS/2020/Workshop/HSDIP — HSDIP 2020_

### Official Review · AnonReviewer2 · 2020-04-01
**Feedback of the Early Assessment Phase**

**Rating:** 7
**Confidence:** 4

**Review:**

This paper shows how algorithms for top-quality planning can be used in order to solve bounded-quality diverse planning problems by generating a set of top quality planners and then selecting a subset of plans that maximize diversity. The latter part is formalized as a MIP.

This is just some feedback for the early assessment phase:


Technical content:

 - One thing that perhaps could be made more clear is how dependent the approach is of
   particular choices for the quality and diversity metrics. In the preliminaries you
   mention that multiple metrics are possible. For example in the diversity metric, we can
   aggregate the results taking the minimum or the average.  In the quality metric, we
   have the one in Definition 1 and then the one in equation (2). If I'm correct, the
   proposed approach works only with the quality metric of equation (2) and the diversity
   metric taking the minimum. Would it be possible to adapt the approach for other
   diversity and quality metrics?


There are several things that would be useful to report in the experiments:

- How many plans are there below the cost bound for each quality bound? I assume that the
  numbers will be completely different for each domain so perhaps you could report the
  number of instances per domain for which |P| <= k, |P| > 10*k, and |P| > 100*k or other
  sensible bounds

- What is the relative time spent in solving the MIP wrt. the time spent finding all plans
  of a given cost?

- I did not see what is the value of k in the analysis of Figure 1.

- In the analysis of Figure 1, you show that the quality of solutions obtained by a greedy
  algorithm is relatively good. However, what is the overall runtime compared to the
  optimal algorithm? If the runtime of the MIP is low compared to that of finding all
  solutions of the right quality bound, then one can always use the FI-bQoptD solver to
  obtain a guaranteed quality in terms of diversity, without much overhead.

 - If the model can be adapted to other quality and diversity metrics, perhaps it would be
   interesting to analyze how this choice impacts the results.x

Related work:

 - You could mention the top-K planner from: "Symbolic Top-K Planning" by David Speck,
   Robert Mattmüller, Bernhard Nebel, recently published at AAAI'20 (e.g. when you say
   that an advantage of your approach is to be able to switch to other top-quality
   planners).



Minor comments:

page 1: While all these planners implement the chosen diversity metric and switching to another metric is not trivial, the planners ->
Most of these ..., except ... which work in two phases


page 5: For the bQbD-k and bQoptD-kcomputational problems -> a space is missing

---

### Official Review · AnonReviewer1 · 2020-04-03
**Some initial feedback**

**Rating:** 8
**Confidence:** 4

**Review:**

The paper is written very clearly. I only have a few minor comments for now:

The abstract could be more concrete and describe which three problems are tackled in the paper.

Definition 1: it's a litte odd to define P as the set of *all known* plans of \Pi. I think it should either be *all* plans, or *known* plans. Also, you could state that the \pi_i are the cheapest plans in P and that n = |P'|. Finally, n' should probably just be n.

The bibtex call should use "-min-crossrefs=99" (or a sufficiently high number) to avoid the standalone AAAI 2020 entry.

---

### Comment · Program_Chairs · 2020-09-14
**Final Decision: Accept**

Dear Authors,

Thank you very much for your submission. We are happy to inform you that we have decided to accept it and we look forward to your talk in the workshop. You will receive additional information per mail in the coming days.

Best,
The HSDIP'20 team

---

### Decision · Program_Chairs · 2020-09-30

Accept